# Prognostic and Predictive Biomarkers in Stage III Melanoma: Current Insights and Clinical Implications

**DOI:** 10.3390/ijms22094561

**Published:** 2021-04-27

**Authors:** Luca Tonella, Valentina Pala, Renata Ponti, Marco Rubatto, Giuseppe Gallo, Luca Mastorino, Gianluca Avallone, Martina Merli, Andrea Agostini, Paolo Fava, Luca Bertero, Rebecca Senetta, Simona Osella-Abate, Simone Ribero, Maria Teresa Fierro, Pietro Quaglino

**Affiliations:** 1Department of Medical Sciences, Dermatologic Clinic, University of Turin, 10126 Turin, Italy; valentinapala@live.it (V.P.); renata.ponti@unito.it (R.P.); rubattomarco@gmail.com (M.R.); giuseppegallomd@gmail.com (G.G.); luca.mastorino@edu.unito.it (L.M.); gianluca.avallone@edu.unito.it (G.A.); martina.merli@edu.unito.it (M.M.); andrea.agostini@edu.unito.it (A.A.); fava_paolo@yahoo.it (P.F.); simone.ribero@unito.it (S.R.); mariateresa.fierro@unito.it (M.T.F.); pietro.quaglino@unito.it (P.Q.); 2Department of Oncology, Pathology Unit, University of Turin, 10126 Turin, Italy; luca.bertero@unito.it (L.B.); rebesenetta@gmail.com (R.S.); simona.osellaabate@unito.it (S.O.-A.)

**Keywords:** melanoma, stage III, biomarkers, adjuvant therapy

## Abstract

Melanoma is one of the most aggressive skin cancers. The 5-year survival rate of stage III melanoma patients ranges from 93% (IIIA) to 32% (IIID) with a high risk of recurrence after complete surgery. The introduction of target and immune therapies has dramatically improved the overall survival, but the identification of patients with a high risk of relapse who will benefit from adjuvant therapy and the determination of the best treatment choice remain crucial. Currently, patient prognosis is based on clinico-pathological features, highlighting the urgent need of predictive and prognostic markers to improve patient management. In recent years, many groups have focused their attention on identifying molecular biomarkers with prognostic and predictive potential. In this review, we examined the main candidate biomarkers reported in the literature.

## 1. Introduction

Melanoma is one of the most aggressive skin cancers, with an increasing incidence worldwide. Incidence varies between countries, with highest value in Australia, New Zealand, North America, and Europe (respectively: 34.9, 35.8, 13.8, and 10.2 per 100,000 person-years) [1,2]. Well known risk factors for cutaneous melanoma include UV radiation by sun exposure, the presence of melanocytic/dysplastic naevi [3,4], phenotypic characteristics (fair hair, skin and eyes colors, freckles), familiar, personal history of cutaneous melanoma and high socio-economic status [5]. Annually, about 0.7% of cancer-related deaths are due to cutaneous melanoma. The prognosis is favorable for early stages of melanoma and poor for metastatic stage IV. Stage III is defined as the presence of nodal or cutaneous local/satellitoses or in-transit metastases. The recent American Joint Committee on Cancer (AJCC) classification eighth edition modified and improved the previous seventh classification through the identification of four different stage III classes (namely A, B, C, and D) defined on the basis of T and N score [6]. The majority of these patients are disease-free after surgery with significantly different relapse risks. The 5-year survival rate ranges from 93% (IIIA: T1a, T1b, T2a; N1a, N2a) to 32% for those with a thick ulcerated disease (T4b) and N3 positive-nodes (IIID) [7]. The risk of recurrence of melanoma after complete surgery is also high for stage IIB, IIC after proper staging, thus including melanoma patients with a thick primary but negative sentinel node biopsy; in particular, the survival of stage IIC is lower than that of stage IIIA, confirming the prognostic relevance of Breslow thickness. Since the early 1990s, immunotherapy with interferon-α (INF-α) has been used as adjuvant therapy, showing limited benefit in terms of OS and PFS [8]. The benefit from interferon therapy appears to be higher for patients with ulcerated primaries vs. non-ulcerated [9]. The finding of high clinical activity associated with high INF-α doses supported by the study of Kirkwood et al. has not been confirmed by further studies, that report only modest clinical results [10]. In recent years, the development of new treatment approaches such as anti-BRAF targeted therapies and checkpoint inhibitors have largely improved survival rates. Large multi-center randomized trials have documented a relevant role played by both the new targeted therapies (dabrafenib and trametinib, vemurafenib and cobimetinib) and immune checkpoint inhibitors (nivolumab and pembrolizumab) in the adjuvant settings with the significant improvement of the PFS and OS, leading to their approval by regulatory agencies [11,12,13]. Despite the good results obtained in terms of PFS and OS [11,14,15], there is a high need for reproducible, sensitive, and “easy-to-use” biomarkers, to guide the clinical decision-making process. Peripheral blood biomarkers have a good potentiality due to the nature of their collection, and they are less invasive and reproducible compared to those tissue-related counterparts. These and other challenges have prompted the investigation of novel biomarkers that could be used as diagnostic, prognostic, and therapeutic aids. By definition, a predictive factor is a condition or finding that can be used to help in predicting whether a patient will respond to a specific treatment. A prognostic factor instead provides information about the patient outcome, regardless of therapy [16]. In this review, we summarize (Table 1) the most recent findings in the field of melanoma biomarkers, focusing specifically on stage III patients and their relationship with disease outcome during or after targeted/immune therapies.

### 1.1. Conventional Clinico-Pathologic Markers and Staging

Well-known clinico-pathologic features represent classic parameters for melanoma staging and prognosis. Concerning primary tumors, tumor thickness and ulceration are the most powerful predictors of survival. The extent of vascular invasion also significantly impacts outcome, but only in the group of thin melanomas (<1 mm). In the N category, three independent factors have prognostic significance: the number of metastatic nodes, whether nodal metastases were clinically occult or clinically apparent, and the presence or absence of MSI (comprising any satellite, locally recurrent or in transit lesion). It has been shown that the survival of patients with only one involved lymph node is significantly superior (51% at 5 years) compared with patients who have two or more [6,17,18]. Moreover, patients with clinically palpable nodes have shorter survival compared with patients with non-palpable disease. The presence of an immune lymphocyte infiltrate within the primary lesion is associated with an improved prognosis in patients with stage III disease. This feature is a protective factor of survival in melanoma patients. However, it is still difficult to differentiate because it is a dynamic event that starts from tumor infiltration by lymphocytes and then evolves into the histological regression [19,20]. The immune infiltrate of regressed melanoma has been proven to have lower counts of CD25+/CD4+, FOXP3+/CD4+, and PD1+/CD4+ lymphocytes compared to the non-regressed ones. The higher expression of RPS6, TP53, NOTCH1, and ABL1 observed in regressed melanomas could be associated with a more preserved cell cycle control, apoptosis, and proliferation [21,22]. This intriguing finding suggests that the characteristics of the intrinsic immune response play a role in immune surveillance against melanoma. Improving the risk models of patients with stage III melanoma would allow one to ameliorate the treatment choice; however, novel circulating markers are lacking. To date, the only marker which has been incorporated for clinical use is lactate dehydrogenase (LDH) whose elevated serum level is an independent and significant predictor of survival, but only for advanced stages.

### 1.2. Gene Expression

In recent years, the development of high-throughput technologies has allowed cancer research to explore gene expression profiles to identify tumor classes, disease-related genes, and new markers for predicting the clinical outcome [60]. The analysis of gene expression and the identification of a single gene or a signature correlated with patients’ outcome could enable better patient stratification, supporting individualized patient management. Different studies demonstrated the association between mRNA-signatures and stage III melanoma patients’ prognosis [23,25,26,27]. Mann et al. [23] identified a 46-gene expression signature whose presence was predictive of better survival (median survival > 100 months and 10 months for patients with and without the signature, respectively). This signature was characterized by an over-representation of immune response genes and it was validated in other two external stage III melanoma datasets. John et al. [24] performed gene expression profiling in 29 clinically palpable nodes derived from stage IIIB and IIIC melanoma patients including 16 “poor prognosis” and 13 “good prognosis” cases, discovering a 21 gene signature able to predict patients’ outcome. Moreover, Bogunovic et al. [25] performed a gene expression profile from patient metastasis and identified a set of 266 genes significantly associated with post-recurrence survival. They observed that genes positively associated with survival were predominantly immune response-related, highlighting the role of immune surveillance in shaping patients’ outcomes, while genes negatively associated with survival were related to cell proliferation. Furthermore, in a multi-center study, Zager et al. [26] evaluated in an independent cohort of cutaneous melanoma patients the prognostic accuracy of a 31-gene signature previously developed and validated (DecisionDx-Melanoma test), able to predict recurrence-free, distant metastasis-free, and survival rates in stage I and II melanoma [61]. They classified, through gene expression profiling as Class 1 (low risk) and Class 2 (high risk), 523 primary tumors, including 69 IIIA stage and 92 IIIB and IIIC stage, and they confirmed the ability of the test to predict the metastatic risk. Many studies have focused on TYRP1, an enzyme-coding gene involved in the production of melanin. Despite its role, the link between TYRP1 and patient survival and how TYRP1 expression acts on cell behavior is still unclear [62,63,64]. Microarray analysis of melanoma metastasis performed by Journe et al. [27] revealed TYRP1 as the first ranked gene associated with shorter survival. The qPCR analysis for TYRP1 expression in the validation set showed a significant correlation with the TYRP1 level and distant-metastasis free survival and OS. This finding suggests a possible role of TYRP1 as a prognostic marker for stage III melanoma patients and a possible target of a therapy. In accordance, El Hajj et al. [28] observed that high TYRP1 mRNA expression in lymph node metastases from melanoma patients was associated with shorter DFS and OS as well as with high Breslow thickness and the presence of ulceration in primary lesions. A large biomarker companion study of the COMBI-AD phase III trial identified multiple immune signatures able to identify inflamed tumors, i.e., with pre-existing immunity in the tumor tissue. Dummer and colleagues found that the expression of the IFN-γ pathway was a robust prognostic biomarker [13]. To characterize the tumor immune activity, they measured the expression of five key genes (IFNG, CXCL9, CXCL10, CXCL11, and GBP1), developing a high or low IFNγ gene expression signature. This is based on median gene expression, without detecting any difference in terms of mRNA expression between the skin and lymph node or between stage subgroups. The baseline IFNγ signature was strongly prognostic in both groups: patients with a high IFNγ signature had a statistically significant superior relapse-free survival, but not overall survival. Another key observation was that there is no association between baseline genetic alterations and the response to therapy or clinical outcome. This finding supports the idea that melanoma cells harboring alterations that confer resistance to MAPK inhibitors are represented by transient and small sub-clones. Only after therapy-induced selection, sub-clones are enriched, and their resistant phenotype becomes prominent. Recently, several clinical trials in the neoadjuvant setting have been initiated for stage III melanoma. Although the results are encouraging, there is a need to understand who can benefit more from the therapy, with acceptable side effects. Rozeman et al. performed a large analysis of biomarkers in the neoadjuvant setting for stage III melanoma patients [65]. The group identified a gene expression signature of INF-γ, which, independently of TMB, is associated with pathologic response and OS. The group was able to discriminate between patients with a high IFN-γ score/TMB score and a low IFN-γ score/TMB score, who, respectively, showed pathological responses of 100% and 39% after neoadjuvant therapy with ipilimumab plus nivolumab.

### 1.3. Micro-RNA

MicroRNAs (miRNAs) are endogenous non-coding RNA molecules, typically 19–22 nucleotides long. MiRNAs regulate the gene expression of their target genes at the post-transcriptional level, through translational repression and/or cleavage. A single miRNA can regulate many different targets. miRNAs bind to their complementary regions, located in the 3′ untranslated region (UTR) of mRNA, and function to inhibit protein translation. Importantly, miRNA expression profiles differ across cancer cell types and non-neoplastic cells, thus, providing a potential therapeutic avenue. MiRNA expression can be detected using PCR and microarray techniques on resected tumor specimens as well as in blood, since a proportion of miRNAs originating from tumors enter the circulation, allowing for non-invasive detection [66,67]. In recent years, many miRNAs have been shown to have potential clinical relevance: Segura et al. found a six-miRNA signature able to improve the risk stratification of stage III patients, identifying high-risk patients who might benefit from adjuvant therapy. The differential expression of most miRNAs from the predictor signature in the metastatic tissue was also observed in the matched-pair primary tumor tissue, suggesting that the miRNA signature may also play a role in the prognosis of early lesions [29]. Similarly, another group showed that low circulating levels of miR-182-5p, and high miR-199a-5p, miR-877-3p, miR-1228-3p, and miR-3613-5p levels, are associated with a higher melanoma stage at the time of primary tumor excision and may serve to anticipate the detection of micrometastatic regional lymph node disease. Therefore, this pattern of miRNA expression could be evaluated to recommend SLN analysis and surveillance follow-up tests for the detection of clinically occult metastatic dissemination and, consequently, to avoid a delay in the initiation of the currently approved therapies [30]. Besides involvement in the tumor stage, miRNAs have been shown to modulate drug resistance mechanisms to immune check-point inhibitors and BRAF/MEK inhibitors. Huber et al., identified a set of miRNAs involved in the conversion of monocytes into immunosuppressive MDSCs (let-7e, miR-99b, miR-100, miR-125a, miR-125b, miR-146a, miR-146b, miR-155), while other miRNAs seem to interfere with PD-1 (miR-28) or PD-L1 (miR-17-5p) expression at a post-transcriptional level, facilitating resistance to immunotherapy [31]. Moreover, it has been suggested that low levels of miR-579-3p can affect the BRAF/MAPK and MDM2/p53 signaling pathways, resulting in uncontrolled cell proliferation and migration, coupled with inhibition of apoptosis, thus contributing to the development of MAPKi resistance [32].

### 1.4. Circulating Tumour DNA

The term circulating tumor DNA (ctDNA) indicates the fraction of cell-free DNA released by tumor cells into the bloodstream as a result of apoptosis, necrosis, or active release from viable cells [68]. In recent years, ctDNA has received substantial attention because it provides information about tumor heterogeneity and evolution over time in a minimally invasive way. ctDNA could have important clinical implications, in particular for advanced stages of melanoma, where the early assessment of drug response, resistance, and/or tumor progression is of primary importance. In a recent study, Marczynski et al. [33] assessed the presence of the most frequent melanoma mutations, BRAF, NRAS, and TERT, in tumor samples from 19 advanced melanoma patients and tracked the mutations in plasma samples with digital PCR technology. ctDNA was detected in 41% of patients and it was associated with shorter progression-free survival. Lee et al. [34] studied the utility of pre-operative ctDNA as a biomarker for the predictive stratification of high-risk stage III melanoma patients undergoing complete lymph node dissection followed by adjuvant treatment. It was shown that the detection of ctDNA was an independent predictor of survival with a higher significance in patients with stage IIID compared to IIIC. It was also associated with a larger nodal melanoma deposit, a higher number of lymph node involvement and an increase in LDH levels. The identification of stage III melanoma patients with a high risk of relapse was also evaluated through the analysis of pre- and post-operative ctDNA levels in patients who received or did not receive adjuvant therapy [35]. It was observed that 14/18 treated patients did not present detectable ctDNA and did not relapse, while the 4/18 treated patients who presented ctDNA in post-operative plasma samples exhibited clearance of ctDNA upon adjuvant treatment and no one relapsed. A different scenario was observed in the cohort of untreated patients with detectable post-operative ctDNA, where the relapse rate was 100%. Moreover, it was also demonstrated that pre-operative ctDNA levels positively correlated with more-advanced melanoma sub-stages associated with a higher probability of recurrence. Furthermore, R.J. Lee et al. [36] showed an increase in five-year-OS, disease-free interval, and metastasis-free interval in high-risk resected melanoma patients with no detectable ctDNA, highlighting the potential of ctDNA as a predictive biomarker of relapse and survival. In a recent study, Gandini et al. [37] performed a systematic review of published articles and meta-analyses to summarize the association between ctDNA and survival in a total of over 2000 stage III and IV melanoma cases. They observed that patients with detectable ctDNA before treatment and during FU had worse PFS and OS, with no differences across tumor stages and systemic therapies. Despite the high potential of ctDNA as a prognostic biomarker, the standardization of a highly sensitive and reproducible methodology is warranted before translating liquid biopsy in clinical practice.

### 1.5. Circulating Tumour Cells

Circulating tumor cells (CTCs) are cancer cells circulating in the peripheral blood shed from either the primary tumor or its metastases. CTCs were discovered in 1869; since that time, enormous progress has been made in understanding their underlying biology. The epithelial-to-mesenchymal transition (EMT) process is believed to play an important role in CTCs dissemination. The use of quantitative real-time reverse transcriptase polymerase chain reaction (RT-PCR) assay has allowed for the rapid quantitative analysis for CTCs detection using both single and multimarker quantitative RT-PCR, although few studies reported this procedure as a predictive surrogate for treatment outcome in stage III melanoma patients. Koyanagi et al. [38] developed a multimarker RT-qPCR assay using four primary and metastatic melanoma markers, MART-1, GalNAc-T, PAX-3 and MAGE-A3, for CTC detection in patients receiving neoadjuvant biochemotherapy (BC) for melanoma. The authors showed that marker detection after overall treatment was associated with significant decreases in relapse-free and overall survival. By multivariate analysis using a Cox proportional-hazards model, the number of markers detected after treatment was a significant independent prognostic factor for overall survival, suggesting that serial monitoring of CTCs could be useful for systemic subclinical disease evaluation. This allows for the molecular evaluation of tumor cell shedding during treatment and for the patient’s outcome prediction after neoadjuvant BC. Consistently, Hoshimoto et al. [39] aimed to identify high risk patients from 320 stage III melanoma patients who were clinically disease-free after complete lymphadenectomy by multimarker RT-qPCR assessment of CTCs. They selected three informative biomarkers (MART1, MAGE-A3, and GalNAc-T) and demonstrated that two or more positive biomarkers were significantly associated with worse distant metastasis disease-free survival and reduced recurrence-free survival. The last decade’s advances in the molecular analysis of miRNA, long non-coding RNA (lncRNA) and ctDNA isolated from the patient blood has led to a golden era of liquid biopsy, reigniting the interest in CTCs. It has been largely shown that the presence of CTCs into the bloodstream has a prognostic value, with an increased risk of recurrence and poor prognosis in many cancers. However, the rare frequency (1–10 CTCs in 8 mL of blood) and heterogeneous expression of specific markers are the main factors limiting the in-depth study of these cells in melanoma patients [69]. In order to elucidate the predictive power of circulating tumor cells, Lucci and colleagues studied 243 stage III-node-positive melanoma patients using the CellSearchTM Circulating Melanoma Cell assay based on a single enrichment marker, the anti-CD146 antibodies, along with anti-CD45 and anti-CD34 to exclude lymphocytes and endothelial cells, respectively [40]. The group showed that one or more circulating tumor cells per 7.5 mL of blood can independently predict disease recurrence at 6 months from baseline, as well as in subsequent months of follow-up (up to 54 months). These results provided excellent insights to support the studies of liquid biopsy techniques, in order to identify optimal candidates for adjuvant systemic therapy. The strengths of this study included the use of a semi-automated liquid biopsy technique for CTC detection and the follow-up involving a relatively high number (243) of node-positive melanoma patients. On the other hand, effective checkpoint inhibitors and targeted therapy regimens were not largely used during the study, so the authors were not able to assert if current adjuvant therapies could affect CTC detection and outcome for these patients. Lin and colleagues [41], through the use of microfluidics-based CTC enrichment with the non-epithelial cellular adhesion molecule (FX1) and a panel of known mRNA and DNA melanoma blood biomarkers, analyzed 22 stage III (a/b/c) melanoma patients. They found that beta catenin 1 (CTNNB1) overexpression could serve as a potential CTC biomarker, suggestive of immune surveillance evasion, which supports studies in human melanoma immune escape mechanisms. Altogether, these results highlighted the importance of CTC evaluation and that serial assessments can detect CTC changes during different phases of treatment. This consideration makes CTC analysis a promising method to detect real-time subclinical tumor spreading. Although great advances have been made in CTC isolation, an important issue is a lack of standardization for a CTC detection due to the high heterogeneity among melanoma CTCs. A deep investigation of CTC phenotypes, their prognostic potential as well as their differential pharmacodynamic responses to treatment is needed [69,70].

### 1.6. Methylation

The methylation of DNA is the essential component of epigenetic modifications that regulate gene expression in different, reversible ways. Its aberration is an epigenetic hallmark of melanoma. It can contribute to melanoma development and progression through different mechanisms that impact cellular pathways related to cell cycle, tumor growth, cellular metabolism, and epithelial to mesenchymal transition [71]. The importance of DNA methylation as a prognostic biomarker in stage III melanoma was studied by Sigalotti et al. [42], who evaluated the genome-wide methylation profiles of short-term neoplastic cell cultures from 45 patients.

Based on global methylation, the K-means clustering algorithm allowed for the classification of the cohort into a favorable group with a median survival of 31.5 months and an unfavorable group with a median survival of 10.3 months, with a 5-year overall survival of 41.2 and 0%, respectively. A 17-gene methylation signature has been identified as sufficient to recapitulate the overall level of genomic methylation and able to distinguish the good prognosis group characterized by low methylation density and the bad prognosis group with high methylation density. Sigalotti et al. [43] investigated the role of global DNA methylation in stage III melanoma by analyzing “long interspersed nucleotide element-1” (LINE-1) methylation levels of short-term tumor cell cultures from patients with nodal disease. It was observed that the 5-year overall survival of patients with hypo-methylated LINE-1 was 48%, compared to 7% for hypermethylated sequences, suggesting that LINE-1 hypomethylation was a significant predictor of increased OS. This result was in contrast with that found by Hoshimoto et al. [44], who evidenced an association between tumor and serum unmethylated LINE-1 level and melanoma progression. The potential of methylation as a prognostic biomarker was confirmed by another study by Tanemura et al. [45], which showed the correlation between high levels of MINT31 methylation and better disease-free survival (DFS) and OS in stage III melanoma. Moreover, Guadagni et al. [46] analyzed with MS-MLPA technology the methylation status of MGMT promoter, known to improve the effect of alkylating agents, to identify melanoma patients with locoregional lesions located in the pelvis, who would benefit from the melphalan regional chemotherapy. Following the examination of 27 metastases, they observed that high levels of MGMT methylation (promoter methylation cut off ≥14%) were associated with longer overall survival in patients treated with melphalan locoregional therapy.

## 2. BRAF Mutation

BRAF is a serine/threonine protein kinase that activates the MAP kinase/ERK-signaling pathway. Somatic oncogenic mutations of BRAF are reported in approximately 50% of melanomas. The substitution of glutamic acid for valine (BRAFV600E) is the most frequent alteration detectable in over 90% of cases, causing the constitutive activation of the kinase as well as insensitivity to negative feedback mechanisms [72,73]. Mutated BRAF is involved in different mechanisms of melanoma progression, including the evasion of senescence and apoptosis, support of the replicative potential, sustained angiogenesis, tissue invasion and metastasis as well as the evasion of the immune response. Many studies evaluated the prognostic significance of BRAF mutations but its role in predicting patient outcome in melanoma is controversial. The majority of studies found an association of BRAF mutation with poor clinical outcome [23,47,48,49]. Mann et al. [23] demonstrated that the absence of BRAF mutation was a favorable prognostic factor in melanoma patients with surgically resected macroscopic nodal metastasis. Barbour et al. [47] analyzed stage IIIB and IIIC melanoma patients who underwent lymph node dissection without neoadjuvant therapy. They observed that patients with BRAF mutations presented higher 3-year recurrence (77%) compared to BRAF wild-type patients (54%) and that locoregional recurrence rarely arose in isolation, highlighting the potential of adjuvant target therapy for this type of patients. Picard et al. [48] also assessed the prognostic power of BRAF mutation in 72 patients with sentinel lymph node dissection in a retrospective study. After testing BRAF status in primary melanoma and lymph node samples, they demonstrated that BRAF mutation was associated with a 4.5-fold higher risk of death compared to the wild-type group, suggesting a notable role of the kinase in tumor spread. Furthermore, in a study by Moreau et al. [49] it was shown that BRAF-mutated melanoma patients with metastatic lymph nodes resection had worse OS and distant-metastasis free survival compared to the wild-type group. Data analysis of BRAF status in patients enrolled in phase III Keynote 054 trial (adjuvant pembrolizumab versus placebo) revealed a different disease outcome in the placebo group with shorter 3-year RFS for BRAF-mutated versus BRAF wild-type melanomas, whilst no differences were found in the pembrolizumab-treated arm [50]. On the other hand, other studies suggested an opposite correlation between BRAF mutation and patient outcome prediction. Tas and Ertuk [51] analyzed the prognostic significance of BRAF V600E mutation in 151 stage III patients; a BRAF mutation was present in 51% of melanomas and was associated with better OS and longer disease-free survival. Consistent results were obtained in a German study by Heppt et al. [52]: melanoma BRAF mutated patients trended towards better overall and melanoma-specific survival. Furthermore, other authors failed to identify any evidence of association between BRAF mutation and survival in stage III melanoma patients [58]. Recent studies have also evaluated BRAF mutation status in extracellular vesicles (EVs). Zocco et al. [74] investigated whether extracellular vesicle-(EV)-associated-DNA (EV-DNA) could be used as an alternative source for assessing circulating BRAFV600E. Using a clinical practice-compatible protocol for the isolation of EV-DNA, they assessed BRAF mutation on plasma samples from metastatic melanoma patients at the beginning and during BRAFi therapy. They found that their proposed protocol improves the detection of BRAFV600E gene copies in comparison to the reference protocol for ctDNA isolation. Moreover, Garcia Silva et al. [75], in an elegant experiment, analyzed EV derived from exudative seroma (ES), a biofluid enriched in EVs, compared with plasma, and demonstrated that ES-EV may represent a useful surrogate marker of melanoma progression and could be used to detect melanoma-specific mutations. Taken together, these findings suggest that EVs could be a promising source of mutant DNA and should be considered for the development of next-generation liquid biopsy approaches.

## 3. Protein Expression

The advent of new technologies has made it possible to study protein expression on a large scale, although the problem is having standard measurement methodologies and a large consensus among clinicians. Mactier et al. [53], in a large-scale proteomic analysis of stage IIIc melanoma patients, found for the first time a signature of 21 proteins able to classify stage IIIc patients into prognostic subgroups (*p* < 0.02). Poor prognosis patients are characterized by increased levels of proteins involved in angiogenesis, methylation, protein metabolism, nucleic acid metabolism, and deregulation of cellular energetics. On the other hand, decreased levels of proteins are involved in apoptosis and immune response. Despite encouraging results from large-scale studies, a wide component of evidence still relies on the study of a limited numbers of proteins, such as S100 proteins. The name S100 refers to the 100% solubility of these proteins in ammonium sulphate, at neutral pH. To date, S100B protein is currently mainly used as an immunohistochemistry marker to confirm melanoma diagnosis in pathological specimens [54]. Karonidis et al. found that serum levels of S100B change according to lymph node involvement in stage III melanoma. Higher levels of S100B were found in N2 (*p* = 0.012) and N3 (*p* = 0.009) compared to N1, while no difference between stages N2 and N3 was detected (*p* = 1.000). Moreover, no correlation was found between the number of primary melanoma lesions and S100B. Wagner et al. [55] measured serum levels of S100A8/A9 and correlated them to survival in a large study, including two cohorts of stage III and stage IV patients. They found that patients who present serum level of S100A8/A9 above 5.5 mg/L have an impaired OS. Combinatory analysis of S100B and LDH each in combination with S100A8/A9 showed a synergistic effect and demonstrated the additional discriminatory power of S100A8/A9 independently of the S100B or LDH levels. Multivariate analysis revealed that S100A8/A9 and S100B, but not LDH, were the only serum markers that independently predicted OS in stage III melanoma. Immune checkpoint inhibitors targeting programmed cell death 1 (PD-1) activate tumor-specific immunity and have shown remarkable efficacy in the treatment of melanoma. It is well known how melanoma-specific PD-1 overexpression enhances tumorigenicity, whereas melanoma-PD-L1 inhibition attenuates the growth of PD-1-positive melanomas [76]. To evaluate PD-L1 status in patients with stage III melanoma, Madore et al. [56] assessed its expression by IHC in 52 AJCC stage III melanoma lymph node specimens and compared these results with specimen-matched comprehensive clinicopathologic, genomic, and transcriptomic data. The results showed that PD-L1-negative status was associated with lower non-synonymous mutation (NSM) burden and worse melanoma-specific survival. Moreover, they identified through gene set enrichment analysis an immune-related gene expression signature in PD-L1-positive tumors with an increase in cytotoxic T-cell and macrophage-specific genes. Weber and Ascierto, in a large multi-center trial, performed a biomarkers analysis in patients with stage III/IV melanoma treated with adjuvant nivolumab vs. ipilimumab. High levels of all the evaluated parameters (interferon-gamma gene expression signature, tumor mutational burden, and CD8+ T-cell infiltration by IHC) showed an association with improved RFS for both nivolumab and ipilimumab. The median RFS observed in nivolumab-treated patients with high vs. low values for each biomarker were 30.8 vs. 24.1, not reached (NR) vs. 30.8, and 30.8 vs. 24.9, respectively; while for ipilimumab, they were NR vs. 15.9, NR vs. 18.3, and NR vs. 13.8, respectively [12].

One of the immune escape mechanisms of cancer is the upregulation of T-regulatory lymphocytes (Treg). Gambichler and colleagues [77] have studied the effect of adjuvant nivolumab on circulating Tregs subpopulations in patients with stage III melanoma. They demonstrated how circulating PD-1 + Tregs rapidly and continuously declined at the beginning of treatment, and CTLA-4+ Tregs levels rose. A logical conclusion is that a combination of anti-PD-1 and anti-CTLA-4 agents in melanoma could serve as a winning strategy even though the higher toxicity of this combined treatment has to be considered, particularly in an adjuvant setting. Finally, Ekmekcioglu and colleagues [57] found that CD74 expression in melanoma cells strongly correlates with improved OS and RFS in stage III melanoma patients. The functional role of CD74 is not well clarified, even though it is known to function in the molecular processing of MHC II, with a potential role in the anti-tumor immune response. Based on an IHC-based study, CD74 was also detected in tumor-infiltrating lymphocytes, also correlating with a statistically significantly better prognosis. In a large retrospective exploratory analysis, Ascierto et al. [58] evaluated the presence of baseline tumor immune infiltrate in 498 patients, randomized to assume placebo or adjuvant vemurafenib. It was found that the presence of CD8+ T cell infiltration and PD-L1 immune cells at baseline is independently associated with better DFS in patients with fully resected, stage IIC-IIIC BRAF V600-mutated melanoma. Another very interesting field of research is the investigation of predictive markers in terms of drug-related adverse events. In a recent study, Lauwyck and colleagues [59] found how C-reactive protein (CRP) may be of use in predicting immune-related adverse events (irAE) related to adjuvant treatment with immune checkpoint inhibitors. Through retrospective analysis of 72 melanoma patients, they observed how in patients that experience irAE, the median serum CRP-levels exceeded the ULN (5 mg/L). Declining CRP-levels were correlated with recovery from an irAE, while increased CRP-levels indicated the relapse of the irAE. Patients who experienced no irAE were at the highest risk for melanoma relapse, while, within patients diagnosed with an irAE, those with an elevated CRP (>2xULN) were at higher risk for relapse compared to those diagnosed with an irAE and CRP < 2xULN.

## 4. Future Perspectives

An efficient and adequate management of cancer patients is determined by early diagnosis, appropriate therapies and disease monitoring during and after treatment. The development of a “personalized” therapy strongly depends on the knowledge of cancer biology and molecular processes that promote tumor progression (Figure 1). In order to do this, in clinical oncology, tumor samples are analyzed for the identifications of somatic mutations. However, genotyping tumor tissue has important limitations for its invasive procedure and cannot capture the tumor heterogeneity. The use of blood as a source of circulating biomarkers, including ctDNA and CTCs, represents a great clinical promise. Indeed, liquid biopsy is a non-invasive procedure, allowing rapid and repeated sampling, a fundamental characteristic for the close monitoring of treatment response and disease progression. Moreover, the analysis of circulating biomarkers allows one to overcome tumor heterogeneity capturing the entire genetic landscape of tumors, with a consequent improvement of treatment choice [78,79,80]. Despite ctDNA and CTC analysis being challenging, especially because of their extreme dilution in the blood, the development of modern technologies has greatly improved the detection sensitivity. Droplet digital PCR (ddPCR) and NGS technologies are the main approaches used for ctDNA detection in different types of cancer. DdPCR could represent a useful tool for patient monitoring; it is a highly sensitive and inexpensive method but requires prior knowledge of mutations. On the other hand, NGS technology can be used for the initial identification of somatic mutations. This technique provides a genomic profile without a priori information, with the disadvantage of being more expensive and less sensitive than ddPCR [81]. Regarding CTC, the difficulties are amplified because common CTC markers used in CTC enrichment of epithelial tumors are not commonly expressed by melanoma-CTCs, since melanocytes are derived from the neural crest [70]. Nevertheless, it is worth investing time and focusing on the standardization of methods in order to exploit the potential of liquid biopsy in clinical practice. Another emerging and promising technology is radiomics, a multi-step approach consisting in the acquisition of medical images, quantitative data extrapolation and their correlation with different endpoints [82]. Regarding melanoma, recent works highlighted the significant role of radiomics images as predictive biomarkers of immunotherapy response and as an important tool for the improvement of cancer patients’ management [83,84,85,86]. Therefore, a multidisciplinary approach integrating biology with bioinformatics and computational science is fundamental in order to discover novel predictive and prognostic biomarkers with the aim of personalizing the treatment of each patient.

## 5. Conclusions

Melanoma represents the most aggressive and lethal form of skin cancer. Although the introduction of adjuvant/neo-adjuvant therapies has provided a remarkable enhancement in DFS and OS, a relevant proportion of patients experience limited clinical benefits. In this scenario, it is important to identify prognostic and predictive biomarkers to stratify and improve stage III melanoma patients’ management. In recent years, several tissues or serologic markers consisting of single molecules or specific signatures have been investigated to help with the monitoring of patients and prognostication. CTCs can serve as an excellent way to follow the progress of the disease and act early in the case of recurrence. Interferon-based immune signatures have a high predictive value for BRAFi/MEKi, while PD-L1/PD-1 can serve as a prognostic indicator. Many efforts are still to be made to understand the potential of miRNAs, BRAF mutation and methylation status: further studies are needed to validate and include these factors in the daily clinical practice. More effort should be invested towards the development of reliable predictive biomarkers, allowing clinicians to only treat patients predicted to benefit from neo/adjuvant therapy. Additionally, it is important that novel biomarkers will be integrated into the design of future randomized controlled trials, so that they can properly and prospectively validated.

## Figures and Tables

**Figure 1 ijms-22-04561-f001:**
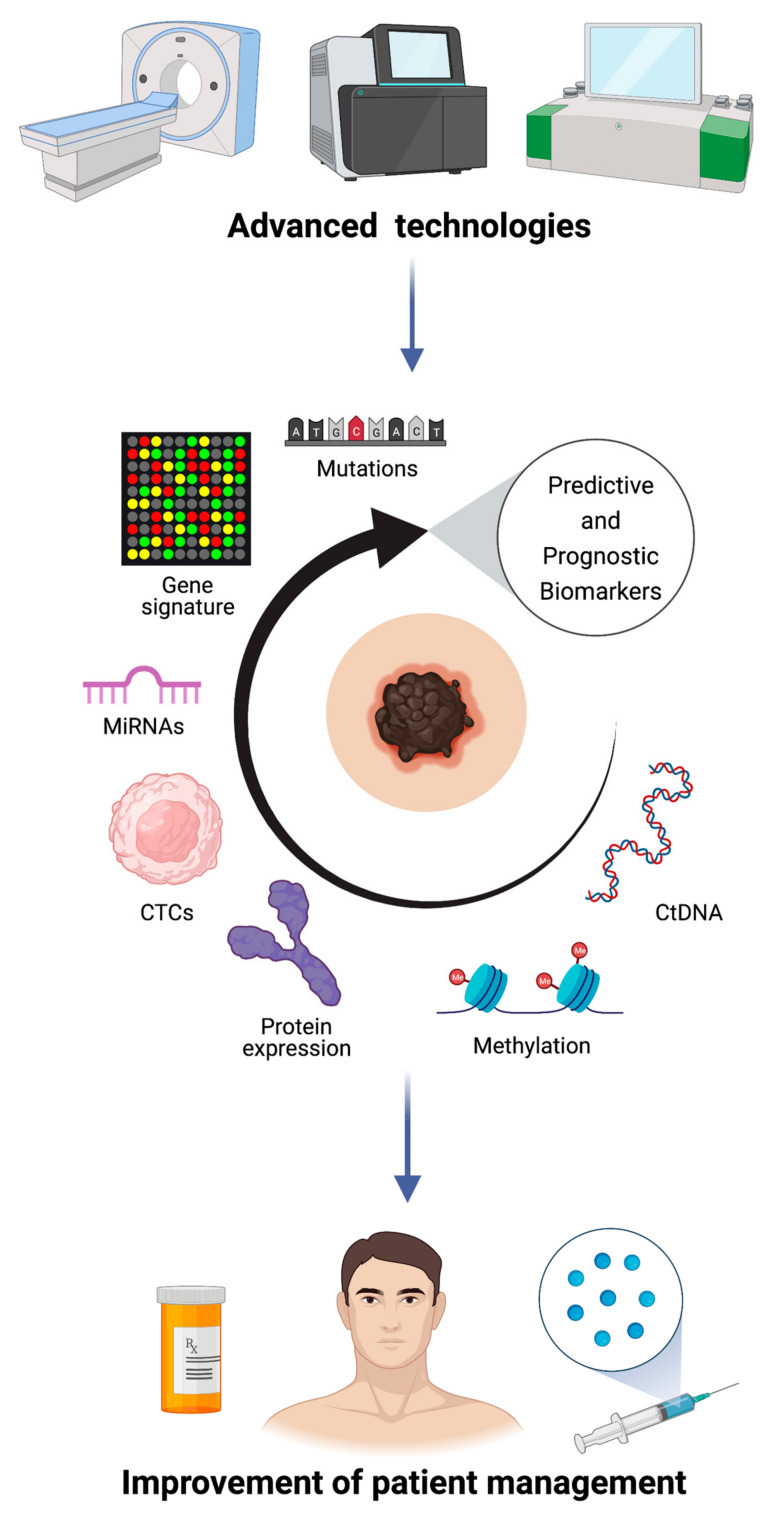
Role of biomarkers in stage III melanoma. Multi-omics approaches are crucial for personalized medicine.

**Table 1 ijms-22-04561-t001:** Summary of the main biomarkers for stage III melanoma.

Type of BM	Author	Findings/Study	Utility OF BMs	Techniques	No. of Patients	Samples
Gene expression	Mann et al. [23]	46 gene expression signature	Prognostic	GEP	79	T + LN
	John et al. [24]	21 gene expression signature	Prognostic	GEP	29	LN
	Bogunovic et al. [25]	266 gene expression signature	Prognostic	Microarray	38	MTS
	Zager et al. [26]	31 gene expression signature	Predictive of metastatic risk	GEP	523	T
	Journe et al. [27]	Expression of TYRP1	Prognostic	Microarray + qPCR	111	T + LN
	El Hajj et al. [28]	Expression of TYRP1	Prognostic	RT-qPCR	104	LN
	Dummer et al. [13]	Expression of IFNG, CXCL9, CXCL10, CXCL11, GBP1	Prognostic and predictive	Nanostring + NGS	875	-
miRNAs	Segura et al. [29]	6 miRNA signature	Prognostic	Microarray	59	LB
	Sanchez-Sendra et al. [30]	5 miRNA signature	Prognostic	RT-qPCR	132	T + LN + MTS
	Huber et al. [31]	MiRNA signature	Predictive of immunotherapy resistance	RT-qPCR	87	LB + T
	Fattore et al. [32]	Expression of miR-579-3p	Predictive of MAPKi resistance	RT-qPCR	23	T
ctDNA	Marczynski et al. [33]	ctDNA (BRAF, NRAS, TERT)	Prognostic	ddPCR	19	LB
	Lee et al. [34]	ctDNA pre-operative	Prognostic	ddPCR	174	LB
	Tan et al. [35]	ctDNA pre- and post-operative	Prognostic	ddPCR	126	LB
	Lee et al. [36]	ctDNA levels	Prognostic	ddPCR	161	LB
	Gandini et al. [37]	ctDNA levels	Prognostic	Meta-analysis	2000	MA
CTCs	Koyanagi et al. [38]	MART-1, GalNAc-T, PAX-3, MAGE-A3 for CTCs detection	Prognostic	RT-qPCR	92	LB
	Hoshimoto et al. [39]	MART1, MAGE-A3, GalNAc-T for CTCs detection	Prognostic	RT-qPCR	320	LB
	Lucci et al. [40]	Anti-CD146 for CTCs detection	Prognostic		243	LB
	Lin et al. [41]	CTNNB1	Predictive	Microfluidics	22	
Methylation	Sigalotti et al. [42]	17 gene methylation signature	Prognostic	Pyrosequencing	42	C
	Sigalotti et al. [43]	LINE-1 methylation levels	Prognostic	BeadChip essay	45	C
	Hoshimoto et al. [44]	LINE-1 methylation levels	Prognostic	MALDI-TOF MS MSP	203	T + MTS + LN + LB
	Tanemura et al. [45]	MINT31 methylation levels	Prognostic	PCR	107	T + MTS
	Guadagni et al. [46]	MGMT promoter methylation levels	Prognostic	PCR	27	MTS
BRAF	Mann et al. [23]	BRAF mutation	Prognostic	GEP	79	T + LN
	Barbour et al. [47]	BRAF mutation	Prognostic	Sequenom MASSarray	134	LN
	Picard et al. [48]	BRAF mutation	Prognostic	PCR	72	T + LN
	Moreau et al. [49]	BRAF mutation	Prognostic	Pyrosequencing	105	T + LN
	Eggermont et al. [50]	BRAF mutation	Prognostic	-	1019	T
	Tas et al. [51]	BRAF mutation	Prognostic	RT-qPCR	151	T
	Heppt et al. [52]	BRAF mutation	Prognostic	Pyrosequencing + Sanger	217	T + LN + MTS
Protein expression	Mactier et al. [53]	21 proteins signature	Prognostic	Mass spectrometry	33	LN
	Karonidis et al. [54]	S100B serum levels	Prognostic	Electroluminescence	107	LB
	Wagner et al. [55]	S100A8/A9 serum levels	Prognostic	ELISA	354	LB
	Madore et al. [56]	PD-L1 expression	Prognostic	IHC	52	LN
	Ekmekcioglu et al. [57]	CD74 expression	Prognostic	IHC	158	LN
	Ascierto et al. [58]	Immune infiltrate	Prognostic	IHC	498	T
	Lauwyck et al. [59]	C-Reactive protein	Predictive of irAEs	-	72	LB

BM: biomarkers, T: primary tumor, MTS: distant metastases, LN: lymph nodes, LB: liquid biopsy, C: cell cultures, MA: meta-analysis, MSP: methylation-specific PCR, MS: mass spectrometry.

## Data Availability

Not Applicable.

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
