# Peer review of "Prognostic and Predictive Biomarkers in Stage III Melanoma: Current Insights and Clinical Implications"

_ijms, 2021, doi:10.3390/ijms22094561_

Round 1

Reviewer 1 Report

This narrative review gives current details of prognostic and predictive biomarkers in melanoma. The article is interesting, and it addresses a current topic, however, there are a few areas that need to be added/improved before publishing.

  • Add summary table of molecular markers associated with diagnosis, prognosis, prediction of therapeutic response of melanoma.
  • Add an illustrative diagram that shows the characteristic, prognostic, and predictive value of markers.
  • Please add a summarizing figure that supports the conclusions section.
  • The author should consider revising the whole manuscript to make sentences simpler in structure for better reader-friendliness. Authors should proofread and correct the grammar mistake as well as sentence structure.

Author Response

Dear reviewer, 

Thank you for your useful suggestions. We think they have improved the quality of our manuscript.

  • We have added a summary table of all biomarkers
  • We have added a figure showing how novel technologies are crucial for biomarkers discovery and application
  • We have revised the manuscript to make senteces more readable

Reviewer 2 Report

The review manuscript by Luca Tonella et al. extensively examined the latest research and findings on predictive and prognostic biomarkers in melanoma. Although technical and specific, the reading of the manuscript appears comprehensible. The authors described various experimental approaches (genomics, proteomics, metabolomics ) for the identification of novel biomarkers able to predict response to treatments. In my opinion, however, it would be necessary to complete the manuscript by also considering the various instrumental methods (e.g. radiomics) which, in recent years, are helping in the clinical and pharmacological evaluation of many forms of cancer, including melanoma.

In addition, the authors could add a short chapter describing the experimental approaches they consider most valuable for the identification of novel biomarkers.

Attention to the layout of the manuscript, the format of the bibliography. Consider the inclusion of summary tables or figures, if appropriate.

Author Response

Dear reviewer,

thank you for your comments, we think they have improved the quality of our manuscript considerably.

  • We have added a new chapter on experimental approaches and new technologies, including radiomics.
  • We have added a summarizing table of all the studies cited
  • We have added a summarizing figure showing the importance of multi-omics approaches for biomarkers discovery and traqnslation into clinical practice
  • We have revised the bibliography

Round 2

Reviewer 2 Report

The manuscript can be accepted in this form